# Development of Novel Follicular Thyroid Cancer Models Which Progress to Poorly Differentiated and Anaplastic Thyroid Cancer

**DOI:** 10.3390/cancers13051094

**Published:** 2021-03-04

**Authors:** Caitlin O. Caperton, Lee Ann Jolly, Nicole Massoll, Andrew J. Bauer, Aime T. Franco

**Affiliations:** 1Department of Physiology and Cell Biology, University of Arkansas for Medical Sciences, Little Rock, AR 72205, USA; cocaperton@uams.edu (C.O.C.); Leeannjolly30@gmail.com (L.A.J.); 2Department of Pathology, Winthrop P. Rockefeller Cancer Institute, University of Arkansas for Medical Sciences, Little Rock, AR 72205, USA; NAMassoll@uams.edu; 3Division of Endocrinology and Diabetes, Department of Pediatrics, The Children’s Hospital of Philadelphia, Philadelphia, PA 19104, USA; BAUERA@email.chop.edu

**Keywords:** thyroid cancer, follicular thyroid cancer, cell lines, MAPK, PI3K, *Hras*

## Abstract

**Simple Summary:**

Advancements in thyroid cancer are dependent on the availability of experimental models allowing for investigation in the laboratory setting. Here, we detail the development and characterization of three cell-based models of follicular thyroid cancer that closely mimic the genetic and pathological progression of the disease seen in patients, including progression to poorly differentiated and anaplastic thyroid cancer in some cases. We anticipate that these newly developed models will allow for the discovery of novel mechanisms of disease progression and the testing of therapeutic interventions.

**Abstract:**

Recent developments in thyroid cancer research have been hindered by a lack of validated in vitro models, allowing for preclinical experimentation and the screening of prospective therapeutics. The goal of this work is to develop and characterize three novel follicular thyroid cancer (FTC) cell lines developed from relevant animal models. These cell lines recapitulate the genetics and histopathological features of FTC, as well as progression to a poorly differentiated state. We demonstrate that these cell lines can be used for a variety of in vitro applications and maintain the potential for in vivo transplantation into immunocompetent hosts. Further, cell lines exhibit differing degrees of dysregulated growth and invasive behavior that may help define mechanisms of pathogenesis underlying the heterogeneity present in the patient population. We believe these novel cell lines will provide powerful tools for investigating the molecular basis of thyroid cancer progression and lead to the development of more personalized diagnostic and treatment strategies.

## 1. Introduction

Thyroid cancer is estimated to have affected 52,890 individuals in the US in the year 2020, making it the most prevalent endocrine malignancy and the most commonly diagnosed cancer among individuals aged 15–29 years [1,2,3]. More than 95% of these cases are comprised of well-differentiated disease, including papillary thyroid cancer (PTC) and follicular thyroid cancer (FTC) [4]. Well-differentiated thyroid cancer has an excellent prognosis with a 5-year survival rate upwards of 90%. A small percentage of well-differentiated cancers are believed to progress and give rise to poorly differentiated thyroid cancer (PDTC) or anaplastic thyroid cancer (ATC). PDTC and ATC are often refractive to chemotherapy and radiation treatment and account for the majority of thyroid cancer-related mortality [5,6,7].

With more conservative diagnostic and treatment recommendations being implemented for well-differentiated disease, the focus has shifted to developing treatments for the more severe PDTC and ATC. Although recent treatment strategies have focused on targeted pathway inhibition [8,9], the success of this approach has been variable. Single-target inhibitors have poor overall response rates, while multiple-kinase inhibitors and treatment with multiple single-target inhibitors have demonstrated more encouraging results [5,9]. Nonetheless, patients often experience cancer relapse within several months of targeted treatment, and toxicity represents a significant challenge when utilizing multi-kinase or multiple single-target inhibitors [10,11].

It has become imperative to develop in vitro and in vivo pre-clinical models that recapitulate patient disease to screen prospective treatments for efficacy and decipher molecular mechanisms that govern therapeutic resistance. Although cell lines derived from human thyroid tumors provide a powerful tool to study novel therapies, progress in the field has been halted by the lack of validated human thyroid cancer lines. In 2008, Schweppe et al. reported that roughly 50% of commonly used thyroid cancer cells lines were misidentified, resulting in cell line redundancy and contamination with cells of other tissue types [12]. Similar misidentification of thyroid cell lines was demonstrated in 2013 [13]. These findings dramatically thwarted progress and led to the re-evaluation of the conclusions from past studies. Many investigators are working to establish new human thyroid cancer cell lines [14,15]. However, research using human lines is limited to the tissue culture environment or xenograft models in immunocompromised hosts, precluding studies aimed at understanding the interactions between tumor cells and the host immune system. In contrast, cell lines established from murine models of thyroid cancer can be transplanted into immunocompetent syngeneic hosts, providing researchers with the ability to study drug responses in the physiological setting of the tumor microenvironment.

In this study, we detail the establishment of three independent cell lines from mouse models of FTC. FTCs are the second most common form of thyroid cancer and approximately half are characterized by activating mutations in *RAS* of the mitogen-activated protein kinase (MAPK) signaling pathway. Further, genetic alterations leading to activation in both the MAPK and phosphoinositide 3-kinase (PI3K)/Akt pathway become increasingly more prevalent as disease progresses, with some estimates indicating that up to 80% of ATCs possess genetic alterations in both pathways [16]. Although FTCs account for a smaller percentage of well-differentiated disease, they possess a unique mode of pathogenesis and progression to PDTC that would benefit from further research [17,18,19]. We have previously described a mouse model of FTC whereby we utilized thyroid-specific expression of *Hras^G12V^* and homozygous *Pten* inactivation in mice to achieve concomitant MAPK and PI3K/Akt pathway activation. These mice develop FTCs that progress to PDTC [*Hras^G12V^/Pten^hom^/TPO-Cre* mice] [20]. This model recapitulates the genetics, histopathological features, and patterns of metastasis of FTC as well as the progression to a poorly differentiated state. Here, we describe the establishment and characterization of three independent cell lines from thyroid tumors of *Hras^G12V^/Pten^hom^/TPO-Cre* mice. These cell lines represent novel and physiologically relevant research tools that can be used for the development of treatment strategies for follicular thyroid cancer, as well as illuminate factors that impact the progression of disease.

## 2. Materials and Methods

### 2.1. Derivation of Murine Thyroid Tumor Cell Lines and Wild Type Thyrocyte Cultures

Cell lines were isolated as previously described [16]. Hras1, H340T, and H245T tumor cell lines were established from thyroid tumors of *Hras^G12V^/Pten^−/−^/TPO-Cre* mice of a pure 129/svJ genetic background. Thyroid tumors and wild-type thyroid glands were dissected and minced, followed by digestion in a solution of 1 mg/mL collagenase Type I (Sigma, St. Louis, MO, USA) and 1 mg/mL dispase (Gibco, Waltham, MA, USA) in Hank’s Balanced Salt Solution at 37 °C with gentle shaking for 1.5 h. Following digestion, samples were centrifuged at 1200 rpm for 3 min and resuspended in Ham’s F12 medium (Corning, Glendale, AZ, USA) supplemented with 10% fetal bovine serum (FBS, Gibco), 2 mM L-Glutamine (Gibco), and Penicillin/Streptomycin/Fungizone (Sigma). The samples were then plated into tissue culture flasks and maintained at 37° Celsius in 5% CO_2_. To ensure removal of contaminating stromal cells and outgrowth of the pure tumor cell lines, all cell lines were passaged at least 5 times after plating and then genotyped using primers specific for *Pten* and *Pten* recombination [21]. Cell lines were authenticated using Short Tandem Repeat (STR) DNA profiling (DDC Medical) according to ANSI guidelines (ASN-0002). Ten mouse STR loci were analyzed for each sample. Loci and STR profiling results are listed in Appendix A.

### 2.2. Immunofluorescence

Cells were seeded into 8-chamber culture slides (Millipore, St. Louis, MO, USA) and allowed to attach overnight. The following day, cells were rinsed with ice-cold PBS and fixed with 4% paraformaldehyde for 10 min at room temperature followed by permeabilization with 0.5% Triton X-100. The cells were treated with 10% goat serum for 1 h prior to antibody staining to block any non-specific binding, and then incubated with anti-EpCAM antibody (1:200, Abcam, Cambridge, MA, USA). The cells were then washed with cold PBS three times for 3 min each and incubated with AlexaFluor 594-labeled secondary antibody (1:200, Invitrogen, Waltham, MA, USA) at room temperature for 1 h. Slides were mounted in SlowFade mounting medium containing 4′,6-diamidino-2-phenylindole (Invitrogen) and imaged using the EVOS FL Auto Cell Imaging System.

### 2.3. RT-PCR Analysis

Total RNA from all cell lines was extracted using the RNeasy Plus Mini Kit (Qiagen, Germantown, MD, USA). Equal amounts of RNA template were reverse transcribed using the Verso cDNA synthesis kit (Thermo Scientific, Waltham, MA, USA). The differential mRNA expression of *Pax8*, *Ttf1*, *Ttf2*, *Tg*, and *Tshr* was measured using pre-designed primers (Integrated DNA Technologies, Coralville, IA, USA, primer sequences listed in Appendix A) and ABsolute qPCR SYBR green mastermix (Thermo Scientific). Four µL of cDNA from tumor samples and independent passages of each cell line were run in triplicate on a Bio-Rad CFX96. *B2m* was used as an internal control in order to normalize the expression levels of thyroid specific genes. Q-Gene software [22] was used to determine relative normalized expression to *B2m*. Data analysis was based on the Ct method.

### 2.4. Western Blot Analysis

Cells were homogenized on ice in RIPA buffer (50 mM Tris-HCL pH 7.4, 1%NP-40, 0.25% Na-deoxycholate, 150 mM NaCl, 1 mM EDTA) supplemented with protease and phosphatase inhibitors (Thermo Scientific). Western blot analysis was performed on 30 µg of total protein separated by SDS-10% PAGE. The proteins were transferred to PVDF membranes and then probed with antibodies (1:1000) specific for PTEN (Millipore), P-AKTS473 (Abcam) pan-AKT, P-ERK1/2, ERK1/2, P-MEK, MEK, and β-actin (Cell Signaling, Danvers, MA, USA). Specific immunoreaction was detected using HRP-conjugated secondary antibodies and Luminata Forte western chemiluminescence substrate (Millipore).

### 2.5. Proliferation Assays

Cells were plated in quadruplicate into 96-well plates. Pharmacologic inhibitors of PI3K (LY294002, 10 µmol/L) and MAPK/extracellular signal-related kinase (ERK) kinase (MEK) (AZD6244, 250 nmol/L) or vehicle (DMSO) were added 24 h after plating. At designated time points, cell growth was assessed using the CellTiter-Glo luminescent cell viability assay (Promega, Madison, WI, USA). Results of cell proliferation in control media were confirmed by plating cells in triplicate in 6-well plates and determining raw cell number via automated trypan blue dye exclusion and the Via XR Cell Viability Analyzer (Beckman Coulter, Brea, CA, USA).

### 2.6. Cytotoxicity Assay

Cells were seeded in triplicate in 96-well dishes. As in proliferation assays, LY294002 (10 µmol/L), AZD6244 (250 nmol/L), or vehicle (DMSO) were added 24 h after plating. At designated time points, media were collected from cells and lactate dehydrogenase release was determined using the CytoTox 96 Non-Radioactive Cytotoxicity Assay (Promega).

### 2.7. Cell Cycle Analysis

Cells were seeded in triplicate in 6-well plates. LY294002 (10 µmol/L), AZD6244 (250 nmol/L), or vehicle (DMSO) were added 24 h after plating. At designated time points, cells and growth media were collected together for analysis. Samples were centrifuged at 2500 rpm followed by washing with PBS. Fixation was carried out with ice-cold 70% ethanol added dropwise while vortexing. Samples were stored in ethanol at −20 °C until staining. On the day of FACS analysis, cells were centrifuged to remove ethanol and stained with 300 µL of propidium iodide/RNAse staining buffer (BD Pharmingen) for 1 h at 4 °C. Samples were analyzed on the LRSFortessa flow cytometer (BD). Data were analyzed using Flow Jo v10 (BD Life Sciences, Franklin Lakes, NJ, USA) and the included Watson model of cell cycle [23].

### 2.8. Three Dimensional Matrigel Growth Assay

Cells were seeded in triplicate into 16-well chamber slides (Millipore) in a 1:1 mixture of Matrigel (Corning) and growth medium at a concentration of 250 cells/well. Thyrospheres were grown for 7 days. Subsequently, samples were rinsed with warmed PBS and fixed with 2% paraformaldehyde for 10 min at room temperature. Cells were permeabilized with 0.1% Triton X-100 and blocked with 1% BSA at room temperature for 1 h. Samples were stained with anti-β-actin primary antibody (1:500, Cell Signaling) for two hours at room temperature or anti-phalloidin antibody conjugated with AlexaFluor 488 (1:40, Invitrogen) for one hour at room temperature. β-actin-stained samples were subsequently incubated with AlexaFluor 594 secondary antibody (1:200, Invitrogen) and mounted in SlowFade mounting medium containing 4′,6-diamidino-2- phenylindole (Invitrogen). Phalloidin-stained samples were mounted in SlowFade mounting medium without DAPI (Invitrogen). All samples were imaged using the EVOS FL Auto Cell Imaging System.

### 2.9. Hanging Drop Assay

Cell suspensions were prepared at a concentration of 2000 cells per 20 µL of growth medium. Cells were pipetted into multiple 20 µL droplets onto the underside of a 10 cm tissue culture dish lid. The lids were carefully inverted and placed onto tissue culture dishes. Culture dishes contained 5 mL of growth medium to prevent the evaporation of droplets. After 7 days, the droplets were removed from the lids by gently washing with PBS and collected in a 6-well culture plate. Viability was assessed via staining with 1 µM of Calcein AM (Life Technologies, Carlsbad, CA, USA) for 30 min at 37 °C. Spheroids were visualized with the EVOS FL Auto Cell Imaging System.

### 2.10. In Vivo Tumorigenecity Assay

All animal experiments were performed at the University of Arkansas for Medical Sciences and approved by the IACUC. Briefly, 5.0 × 10^5^
*Hras^G12V^/Pten^−/−^/TPO-Cre* tumor cells (Hras1, H245T, H340T) were pelleted by centrifugation, resuspended in 200 μL of a 1:1 mixture of PBS and matrigel (Corning), and injected subcutaneously into the right hind flanks of 8–12-week-old male wild-type 129/SvJ recipients. Tumor development was monitored weekly for 15 weeks, at which point the mice were sacrificed due to tumor burden.

### 2.11. FACS Analysis

Tumors from *Hras^G12V^/Pten^−/−^/TPO-Cre* mice were dissected, minced and digested in a solution of serum-free DMEM (Gibco) containing 1 mg/mL Collagenase I (Sigma) and 1 mg/mL Dispase (Sigma) for 1.5 h at 37 °Celsius with gentle shaking. The digested samples were then filtered through 70μm nylon strainers (Fisher Scientific, Hampton, NH, USA) into 10 cm tissue culture dishes containing 10 mL of PBS. The samples were treated with 100 ug/mL DNAse I (Sigma) for 5 min at room temperature, centrifuged for 5 min at 1400 rpm, and resuspended in 1 mL red-cell lysis buffer to remove residual red blood cells. After 5 min of incubation at room temperature, the red-cell lysis buffer reaction was neutralized with 20 mL of PBS. The cell suspensions were filtered again through 70 μM nylon strainers, centrifuged, and resuspended in FACS buffer (1% BSA, 2 mM EDTA in PBS). Cell suspensions were treated with Fc receptor blocking antibody (BD biosciences) for 20 min on ice. After Fc receptor blocking, antibodies specific to the following immune cell populations were added to the cell suspensions and incubated for 25 min at 4 °C in the dark: CD45-PECy7 (1:100, Biolegend, San Diego, CA, USA), CD11b-PE (1:100, Abcam), Gr-1-APC (1:100, BD Biosciences), F4/80-FITC (1:100, Biolegend). The samples were then washed with FACS buffer, centrifuged, and resuspended in FACS buffer containing Zombie Aqua viability dye (Biolegend). Samples were immediately analyzed on the LRSFortessa flow cytometer (BD). The data were analyzed using Flow Jo V.10 (BD). Samples were gated for immune cells of the myeloid (CD45+, CD11b+) or lymphoid lineage (CD45+, CD11b−). Myeloid lineage cells were further divided into macrophages (F4/80+, Gr1−) and neutrophils (F4/80−, Gr1+).

### 2.12. Conditioned Media Collection

*Hras^G12V^/Pten^−/−^/TPO-Cre* cells (Hras1, H245T, H340T) were plated at 300,000 cells/well in 6-well dishes in triplicate. In addition, wildtype thyrocyte cultures were generated as described above. When cells reached confluency, complete media was replaced by serum-free Ham’s F12 medium (Corning). Conditioned media were collected after 18 h, filtered through 40μm nylon strainers (Fisher Scientific) to remove debris, and stored at −20 °C until analysis.

### 2.13. Proteomics Analysis

Conditioned media samples were processed and analyzed by the UAMS Proteomics Core, as follows: Purified proteins were reduced, alkylated, and digested using filter-aided sample preparation with sequencing grade modified porcine trypsin (Promega). Tryptic peptides were then separated by reverse phase XSelect CSH C18 2.5 um resin (Waters) on an in-line 150 × 0.075 mm column using an UltiMate 3000 RSLCnano system (Thermo). Peptides were eluted using a 60 min gradient from 97:3 to 60:40 buffer A:B ratio (Buffer A: 0.1% formic acid, 0.5% acetonitrile; Buffer B:0.1% formic acid, 99.9% acetonitrile). Eluted peptides were ionized by electrospray (2.15 kV), followed by mass spectrometric analysis on an Orbitrap Fusion Lumos mass spectrometer (Thermo). MS data were acquired using the FTMS analyzer in profile mode at a resolution of 240,000 over a range of 375 to 1500 *m*/*z*. Following HCD activation, MS/MS data were acquired using the ion trap analyzer in centroid mode and normal mass range with precursor mass-dependent normalized collision energy between 28.0 and 31.0. Proteins were identified by database search using MaxQuant (Max Planck Institute, Martinsried, Munich, Germany) with a parent ion tolerance of 3 ppm and a fragment ion tolerance of 0.5 Da. Scaffold Q+S (Proteome Software, Portland, OR, USA) was used to verify MS/MS-based peptide and protein identifications. Protein identifications were accepted if they could be established with less than 1.0% false discovery and contained at least 2 identified peptides. Protein probabilities were assigned by the Protein Prophet algorithm (Anal. Chem. 75: 4646-58 (2003)). Protein intensities were normalized using iBAQ (intensity-based absolute quantification) algorithm and differences between groups were calculated with assistance from the UAMS Bioinformatics Core. All differences between cell lines (Hras1, H245T, H340T) and wildtype secretions with a fold change of greater than 2 and a multiplicity-adjusted *p*-value of ≤0.05 were considered for analysis. REACTOME pathway analysis software was used to identify known pathways containing protein targets. Pathways were ranked by the false discovery rate (FDR) value obtained from over-representation analysis.

### 2.14. Statistical Analysis

Excluding proteomics data, all other data were analyzed using Prism 6 software (GraphPad, San Diego, CA, USA). Differences with *p*-values of ≤0.05 were considered statistically significant. All data comparing more than two groups were analyzed via ANOVA analysis with post-hoc analysis (indicated in figure legends).

## 3. Results

### 3.1. Generation of Stable Tumor Cell Lines from Hras^G12V^/Pten^−/−^/TPO Cre Thyroid Tumors

Follicular carcinomas developing in *Hras^G12V^/Pten^−/−^*/*TPO-Cre* mice were dissected and enzymatically dissociated as previously described (Figure 1A). Cell suspensions were plated in Ham’s F12 medium supplemented with 10% FBS. Homogenous populations of epithelial colonies began to expand over the course of 3–4 passages. A total of three cell lines were established from three individual *Hras^G12V^/Pten^−/−^/TPO-Cre* thyroid tumors (Hras1, H245T, H340T). Hemotoxylin and eosin staining of *Hras^G12V^/Pten^−/−^/TPO-Cre* tumors confirmed the model from which these cell lines were derived exhibit FTC-like histological characteristics, including enlarged follicular cells with mostly solid growth patterns, some intervening follicular patterns and pleomorphic nuclei (Figure 1B). During pathological evaluation, it was noted that the source tumor for cell line H340T was more advanced and exhibited multiple neoplastic foci.

*Hras^G12V^/Pten^−/−^/TPO-Cre* mouse models undergo thyroid epithelium-specific *Pten* recombination via the action of *TPO-Cre*. To confirm the presence of *Pten* recombination in established cell lines, PCR analysis using primers specific for recombined *Pten* alleles was utilized. As expected, each cell line exhibited *Pten* recombination in comparison to tail DNA isolated from the tumor-bearing animal from which the cell line was derived (Figure 1C). Western blot analysis confirmed complete loss of PTEN in all cell lines compared to WT primary thyrocytes (Figure 1D). DNA isolated from each cell line was subjected to STR profiling (DDC Medical). *Hras^G12V^/Pten^−/−^/TPO-Cre* cell lines were generated from independent mice, but on a pure 129/SvJ genetic background; therefore, STR analysis was not sensitive enough to distinguish between the genetically identical STRs of the pure 129/SvJ genetic background (Appendix A). However, this analysis confirmed that the cell lines were of mouse origin and were not contaminated with any human or known cell lines.

### 3.2. Morphological Features and Expression of Thyroid-Specific Genes in Hras^G12V^/Pten^−/−^/TPO Cre Thyroid Tumor Cell Lines

Morphological features of the *Hras*-driven cell lines were established via phase-contrast and immunofluorescence imaging. Hras1, H245T, and H340T exhibited spindle-like cell morphology and failed to grow in epithelial colonies, indicating loss of cell–cell adhesion. (Figure 2A). Cell lines also demonstrated an absence of staining for the epithelial cell adhesion marker (EpCAM) when compared to primary thyrocytes (Figure 2B), which likely contributed to the observed growth pattern. Because advanced thyroid tumors often demonstrate loss of epithelial markers, we hypothesized that lack of EpCAM expression is due to de-differentiation of the tumors from which these cell lines were derived rather than an artifact of cell culture.

Next, we sought to validate the thyroid origin of these cells and their differentiation status by measuring the mRNA levels of thyroid-relevant genes, including *Pax-8, Ttf1, Ttf2*, *Tg* and *Tshr* (Figure 2C). *Pax-8*, *Ttf-1*, and *Ttf-2* are transcription factors crucial to thyroid organogenesis and their absence results in varying degrees of thyroid dysgenesis [24,25]. Loss of all or some of these transcription factors is commonly observed in anaplastic thyroid carcinoma and thus their expression has been investigated for diagnostic purposes [24,26]. *Pax-8* expression levels in H340T and H245T cell lines are comparable to that of wild-type (WT) primary thyrocytes, while an increase in expression occurred in Hras1. The detection of *Pax-8* transcripts confirms the thyroid origin of these cell lines. All three *Hras^G12V^/Pten^−/−^/TPO-Cre* thyroid cell lines (Hras1, H245T, H340T) had detectable expression levels of *Ttf-1.* In contrast, all lines lacked the expression of *Ttf-2*, providing additional evidence of de-differentiation of the tumors from which these cell lines were derived. Collectively, *Pax-8*, *Ttf-1*, and *Ttf-2* regulate the expression of *Tg* and *Tshr* [24]. *Tg* and *Tshr* expression levels were undetectable in all cell lines. Similar to thyroid transcription factors, loss of *Tg* and *Tshr* expression has been associated with disease progression [27,28]. Shweppe et al. and Henderson et al. similarly reported varying expression of thyroid-specific genes in their patient-derived thyroid cancer cell lines [14,15]. Overall, these results coupled with histologic and genetic analysis of Hras^G12V^/Pten^−/−^*TPO-Cre* tumors demonstrate that the Hras-driven cell lines represent a true model of follicular thyroid cancer and its progression to a dedifferentiated state.

### 3.3. Proliferation Rates of Hras^G12V^/Pten^−/−^/TPO-Cre Cell Lines In Vitro

We next determined the growth rates of the cell lines in culture (Figure 3A) using the CellTiterGlo luminescence-based viability assay (Promega) and analysis of raw cell numbers. The CellTiterGlo method is based on the quantification of ATP amounts to determine the number of metabolically active cells. This method demonstrated a lag-phase of approximately 24 h after baseline measurements (Figure 3A, top). Cells had comparable growth rates, with Hras1 exhibiting the fastest rate of growth, and by day 3 cell growth began to decline. Analysis of raw cell numbers similarly demonstrated an initial lag-phase with stabilization of cell number around day 3 (Figure 3A, bottom). However, Hras1 and H245T exhibited roughly equal rates of growth that were not reflected in the ATP-based method of assessing proliferation. These results may indicate slight differences in metabolic activities of the individual cell lines. Nonetheless, Western blot analysis demonstrated that all cell lines had roughly equivalent levels of MAPK activation, particularly in regard to phosphorylated AKT (pAKT), phosphorylated ERK1/2 (pERK1/2) and phosphorylated MEK (pMEK) (Figure 3B).

### 3.4. Growth Suppressive Effect of Pathway-Specific Inhibitors in Hras-Driven Thyroid Tumor Cell Lines

We next evaluated the efficacy of pathway-specific inhibitors on the growth of *Hras^G12V^/Pten^−/−^/TPO-Cre* cell lines by treating cells with LY294002 (PI3K inhibitor) and AZD6244 (MEK inhibitor) alone or in combination for 72 h. In all cell lines, the inhibition of PI3K or MEK alone resulted in a significant reduction growth (Figure 4A), although H245T appeared to be slightly more resistant to PI3K inhibition alone. Dual inhibition of MEK and PI3K resulted in further reductions in growth.

We further attempted to determine whether the effects of PI3K and MEK inhibitors on proliferation were mediated via cytotoxic or cytostatic mechanisms. To determine degree of cytotoxicity, we measured lactate dehydrogenase (LDH) in the supernatant of cells treated with PI3K and MEK inhibitors using a colorimetric assay. In all cell lines, the inhibition of PI3K and MEK alone or in combination caused no increase in cell death relative to vehicle (Figure 4B). Cell cycle analysis demonstrated that cells treated with PI3K or MEK inhibitors experienced G1 arrest and a corresponding reduction in the number of cells undergoing DNA replication in the S phase, particularly during day two and day three of treatment (Figure 4C,D). PI3K inhibition with LY294002 appeared to be the most effective at delaying progression through the cell cycle. Overall, our results demonstrate that the *Hras^G12V^/Pten^−/−^/TPO-Cre* cell lines provide a valuable tool for deciphering the molecular mechanisms of targeted therapies.

### 3.5. Development of Physiologically Relevant Three-Dimensional In Vitro Tumor Models

Although the two-dimensional growth of cells on plastic dishes remains the most popular method of in vitro experimentation, three-dimensional (3D) models of tumor growth are gaining favorability as they better recapitulate the environment of solid tumors and cellular responses to therapeutics. As a result, we sought to develop more physiologically relevant methods of growing our cell lines using previously established 3D culture methods. When grown embedded in Matrigel matrix, staining for actin filaments revealed that Hras1 grew as asymmetric multicellular aggregates with loose cytoskeletal arrangement, while H245T and H340T grew in dense, circular formations. (Figure 5A, top and bottom). Further, H245T produced small spheroids of roughly 50 µm in size, compared to the much larger 200 µm Hras1 and H340T spheroids. We also grew Hras-driven cell lines via the hanging drop method of spheroid formation, wherein cells were suspended in droplets from the lid of a culture dish (Figure 5B, left). Calcein AM viability staining revealed that cells were viable throughout the spheroid, including within the core, for up to 7 days in culture. (Figure 5B, right). In contrast to Matrigel embedded growth, this method generated spheroids that were roughly equal in size across all cell lines, potentially due to the aggregation produced by gravitational forces in suspended droplets. The ability to easily and quickly generate 3D cultures with *Hras^G12V^/Pten^−/−^/TPO-Cre* cell lines offers the potential for rapid screening of therapeutics and more in-depth mechanistic studies in environments that better mimic the in vivo tumor environment.

### 3.6. Development of Syngeneic Hras^G12V^/Pten^−/−^/TPO-Cre Tumor Model

The subcutaneous injection of tumor cells into murine hosts has proved to be a convenient method for exploring tumorigenesis and evaluating the potential effects of anti-cancer agents. The strengths of the subcutaneous tumor model coupled with the long latency of our *de novo Hras^G12V^/Pten^−/−^/TPO-Cre* murine model [20] led us to develop a syngeneic tumor model of FTC. We evaluated the tumorigenic capacity of the *Hras^G12V^/Pten^−/−^/TPO-Cre* cell lines in immunocompetent hosts by injecting cells subcutaneously into the flank of wildtype 129SvJ mice. Mice were monitored weekly for 15 weeks for the development of tumors. Upon sacrifice, 7 of the 11 mice were injected with Hras1 developed tumors, resulting in a tumor penetrance of 64%. Conversely, 5 of the 12 mice (42%) injected with H340 exhibited tumor growth. Mice injected with H245 were observed and monitored for 15 weeks, at which time they were euthanized, and no evidence of any tumor formation was found in these animals (Figure 6A).

Gross examination of tumors that developed in mice injected with H340 or Hras1 revealed extensive neovascularization around developing tumors in comparison to the contralateral hind flank control (Figure 6B). Histopathological analysis revealed multiple mitotic cells as well as pleomorphic nuclei with vesicular changes and prominent nucleoli. In contrast to source tumors for cell line H340T, subcutaneous tumors derived from H340T cells were poorly differentiated or anaplastic with no follicular pattern of growth (Figure 6C). Because FTC commonly spreads to the lungs, bone and brain in the clinical setting [29], we investigated whether H340T or Hras1 would metastasize to these sites from subcutaneous tumors. Gross metastatic pulmonary lesions were observed in roughly half of the Hras1 tumor bearing mice. No gross lung lesions were observed in H340T inoculated mice.

The long latency of *Hras^G12V^/Pten^−/−^/TPO-Cre de novo* tumors to develop in the thyroid precludes many in vivo experiments. We sought to determine whether subcutaneous tumors were able to recruit and remodel a tumor microenvironment similar to that observed in de novo tumors in the thyroid. FACS analysis was carried out in order to determine the cellular composition of subcutaneous tumors relative to the *Hras^G12V^/Pten^−/−^/TPO-Cre* de novo model. We have previously reported that the *Hras^G12V^/Pten^−/−^/TPO-Cre* de novo model demonstrated extensive CD45+ immune cell recruitment to the tumor microenvironment [20]. The subcutaneous model recruited a microenvironment strikingly similar to that of the de novo model, with a similar proportion of CD45+ and CD45- cells (Figure 6D). Similar to the de novo model, the subcutaneous model was enriched with myeloid lineage cells (CD11b+), while lymphoid lineage cells (CD11b-) represented a much smaller proportion of CD45+ immune cells. Within the myeloid lineage, macrophages (F4/80+, Gr1-) were the largest cell population in both models (Figure 6E). In sum, *Hras^G12V^/Pten^−/−^/TPO-Cre* cell lines represent valuable models for in vivo investigation and allow for studying tumor-immune interactions in immunocompetent hosts.

### 3.7. Proteomic Analysis of Conditioned Media from Hras^G12V^/Pten^−/−^/TPO-Cre Cell Lines

Based on the differences in disease latency and tumor penetrance, we hypothesized that *Hras^G12V^/Pten^−/−^/TPO-Cre* cell lines secreted molecules that could be responsible for altering the tumor microenvironment at the cellular level, particularly in regard to immune cell recruitment. To investigate this idea, we performed comprehensive proteomic analysis of conditioned media isolated from all three Hras-driven cell lines (Hras1, H245T, H340T) and compared it to secretions from cultured wildtype thyrocytes (Figure 7). Pathway analysis using REACTOME pathway database revealed that some of the most highly upregulated pathways in Hras1 and H340T were related to immune recruitment, particularly cytokine signaling. Indeed, the immune pathway as a whole contained the largest number of matching proteins secreted by these cell lines. In contrast, there were far fewer proteins secreted from H245T that were statistically significant from wildtype secretions and those proteins were only weakly mapped to an immune-related pathway, potentially indicating a mechanism of pathogenesis differing from the other two cell lines and not based on secreted proteins. Additional experimentation will be needed to determine the biological significance of these trends and the effects of secreted proteins on cancer pathogenesis. However, these results demonstrate the heterogeneity within thyroid cancer cells, and represent a novel tool for elucidating the roll of these differentially secreted factors in tumor progression and immune remodeling.

## 4. Discussion

Cancer cell lines derived from tumors of genetically engineered mouse models represent an invaluable tool that can be used for in vitro and in vivo research and pre-clinical studies for thyroid cancer. Here, we sought to generate cell lines specifically targeted at understanding the pathogenesis of follicular thyroid cancer (FTC). *RAS* mutations are observed in roughly 50% of FTCs and secondary mutations in the PI3K/Akt pathway are also commonly seen [16,30]. In accordance with these genetic trends, we have previously shown that activation of *Hras* and deletion of *Pten* (*Hras^G12V^/Pten^−/−^/TPO-Cre*) led to the development of FTC in mice [20]. Aside from genetic alterations, this model further recapitulates FTC by progressing to poorly differentiated disease as well as by exhibiting hematogenous metastasis and possessing similar histopathologic features. We have subsequently generated and described three independent cell lines that are derived from *Hras^G12V^/Pten^−/−^/TPO-Cre* mice and maintain many of the same characteristics as the de novo murine model. We have shown that these cells lines can be used for a variety of in vitro applications including mechanistic studies of targeted inhibitors and 3D culturing, while also demonstrating successful transplantation into syngeneic hosts. For these experiments, we chose to highlight a subcutaneous model of in vivo tumor growth as it represents a method of exploring tumorigenesis and evaluating the potential effects of anti-cancer agents that is readily accessible to researchers of many backgrounds without extensive training. However, we anticipate these cell lines could be used in more advanced in vivo modeling methods, including orthotopic implantation directly into the thyroid of wildtype recipients. Although beyond the scope of our present study, it would be of interest in future experiments to determine whether tumors developed from orthotopic implantation of the described cells would maintain a more well-differentiated phenotype in the endogenous thyroid environment. Additionally, although these cells were all derived from mice of the same genetic background, our results demonstrate heterogeneity within these cell lines that may better represent differences in clinical behavior observed in the patient population.

Over the last five to ten years, the focus of therapeutics has been heavily directed towards the treatment of radioiodine-refractory, poorly differentiated and anaplastic thyroid cancer. All of the Hras-driven lines described here partially express the transcriptional machinery necessary for normal thyroid differentiation; however, the expression of other markers of thyroid function including *Tg* and *Tshr* are undetectable in all cell lines. This observation is likely reflective of the transition of the original endogenous tumors to a poorly differentiated state in *Hras^G12V^/Pten^−/−^/TPO-Cre* mice. As a result, these cell lines will allow for the investigation of the molecular basis by which thyroid tumors progress to poorly differentiated disease and for the testing of targeted signaling-pathway inhibitors that may potentially restore thyroid function.

## 5. Conclusions

In sum, the establishment of three independent cell lines modeling follicular thyroid cancer represents a step towards developing much needed validated in vitro models for thyroid cancer research. We anticipate that these cell lines will offer powerful tools for in vitro and in vivo studies that will lead to the discovery of more personalized diagnostic and treatment strategies for patients with refractory thyroid cancer.

## Figures and Tables

**Figure 1 cancers-13-01094-f001:**
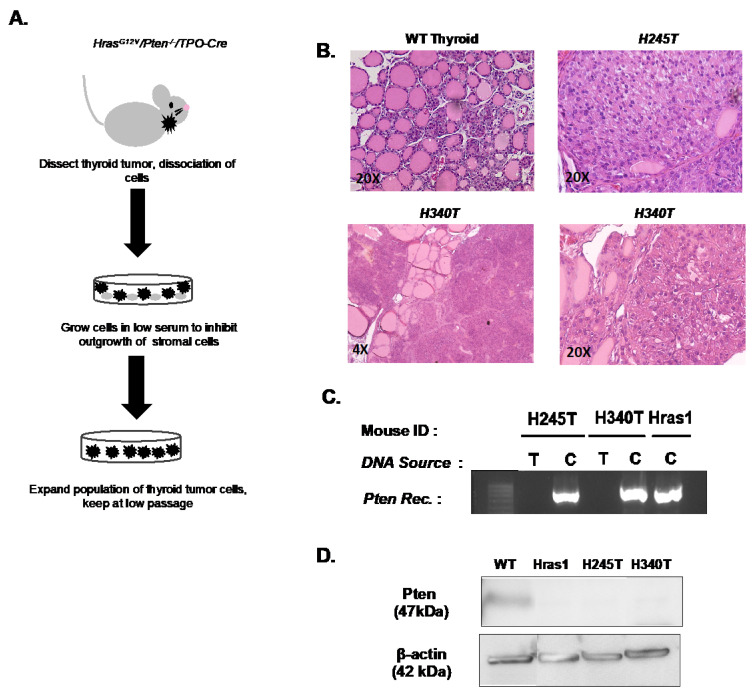
Generation of stable tumor cell lines from *Hras^G12V^/Pten^−/−^/TPO-Cre* thyroid tumors. (**A**) Schematic of thyroid tumor cell line derivation from Hras^G12V^/Pten^−/−^/*TPO-Cre* mice. (**B**) Representative hematoxylin and eosin staining of normal thyroid histology and source tumors for cell lines H245T and H340T. Compared to WT thyroid, H245T and H340T tumors demonstrated mostly solid growth with some intervening follicular patterns, enlarged follicular cells, and pleomorphic nuclei (20× images). H340T source tumor was more advanced with multifocal neoplastic nodules (4× image). Histology of source tumor for Hras1 cell line is not available. (**C**) Polymerase Chain reaction (PCR) analysis of matched pairs of control tail DNA (T) and thyroid tumor cell line DNA (**D**) Western blot analysis of Pten expression in all cell lines compared to primary wild-type (WT) thyrocytes. Original blots see Appendix A

**Figure 2 cancers-13-01094-f002:**
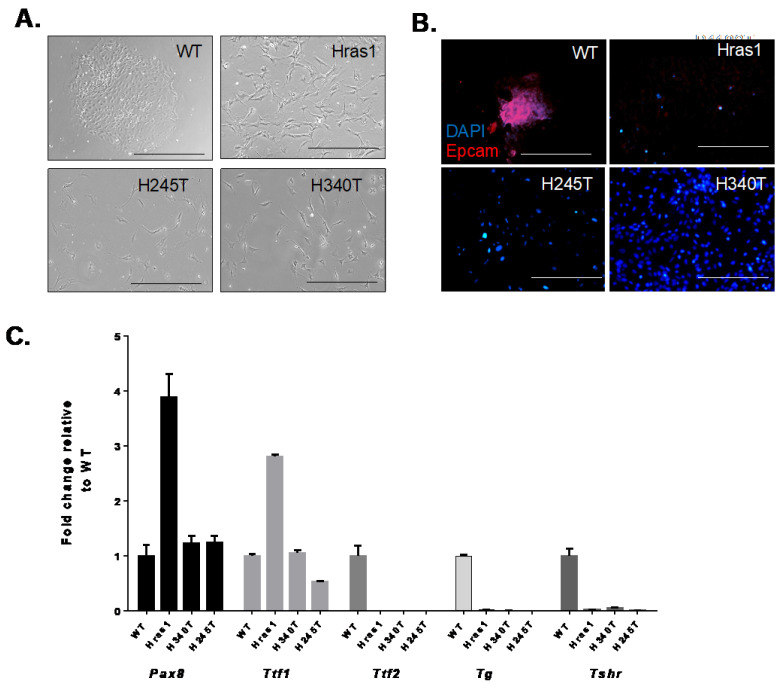
Morphological features and expression of thyroid-specific genes in *Hras^G12V^/Pten^−/−^/TPO-Cre* (Hras1, H245T, and H340T) thyroid tumor cell lines. (**A**) Phase-contrast images (10×) of primary wild-type (WT) thyrocytes and thyroid tumor cell lines in culture. Scale bar represents 400 μm. (**B**) Fluorescent imaging (20×) of EpCAM immunostained WT (positive control) thyrocytes and thyroid tumor cell lines in culture. Scale bar represents 200μm. Nuclei are stained with 4′, 6-diamidino-2-phenylindole (DAPI). (**C**) Real-time PCR analysis of the expression levels of thyroid-specific genes in primary wild-type (WT) thyrocytes and thyroid tumor cell lines in culture. Data are representative of three independent experiments. Each sample was analyzed in triplicate (*n* = 3).

**Figure 3 cancers-13-01094-f003:**
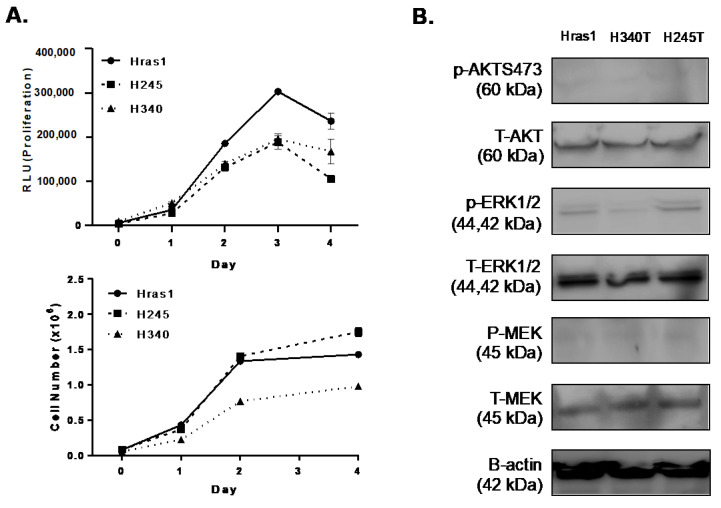
Proliferation rates of thyroid tumor cell lines and activation of MAPK pathway signaling. (**A**) Growth curves of *Hras^G12V^/Pten^−/−^/TPO-Cre* thyroid tumor cell lines (Hras1, H245T, and H340T)*. Top:* Measurement of proliferation using the Cell-Titer Glo proliferation assay. *Bottom:* Analysis of proliferation via raw cell counts. Day 0 represents 18 h after plating equal numbers of cells. (**B**) Western blot analysis confirming activation of MAPK pathway signaling in thyroid tumor cell lines. For all data, representative experiments out of 3 are shown. Each sample was analyzed in triplicate (*n* = 3). Original blots see Appendix A

**Figure 4 cancers-13-01094-f004:**
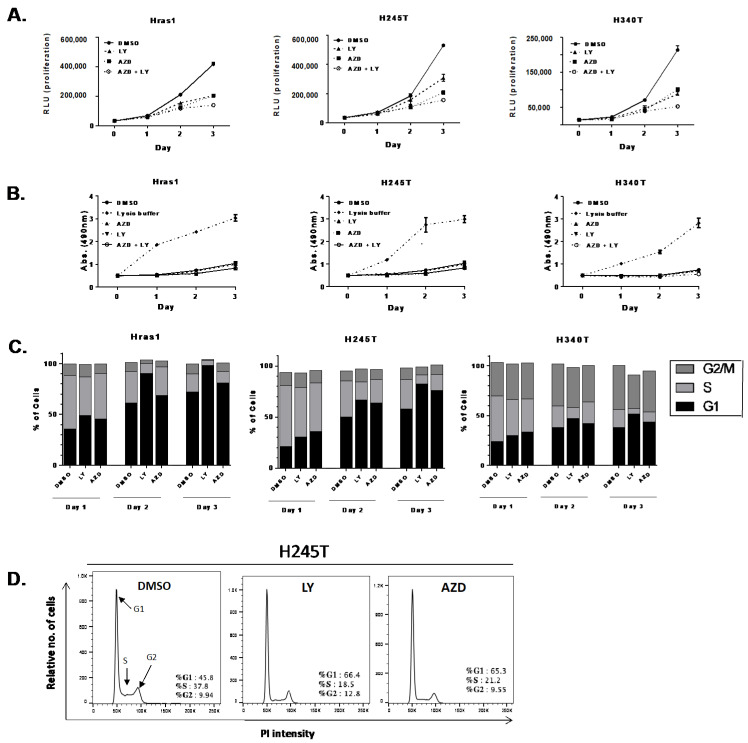
Growth suppressive effects of MEK and PI3K inhibition in *Hras^G12V^/Pten^−/−^/TPO Cre* cell lines. (**A**) Growth curves of thyroid tumor cell lines (Hras1, H245T, and H340T) treated with MEK inhibitor (AZD:AZD6244), PI3K inhibitor (LY: LY294002) or combination of MEK and PI3K inhibitors (AZD + LY). Day 0 represents 18 h after plating equal numbers of cells. Results compared to treatment with vehicle alone (DMSO). (**B**) Analysis of cell cytotoxicity in thyroid tumor cell lines via colorimetric detection of lactate dehydrogenase (LDH) in cell supernatant. Cells were treated with MEK and PI3K inhibitors for 72 h prior to analysis. Lysis buffer treatment simulates maximum potential LDH release at given measurement time point. (**C**) Summary cell cycle analysis results of cells treated with MEK and PI3K inhibitors for 72 h. Results demonstrate relative DNA content and corresponding percentage of cells in G1, S, and G2/M phase. (**D**) Representative FACS analysis of propidium iodine staining in H245T cell line treated with MEK and PI3K inhibitors. Results demonstrate a decrease in the relative number of cells in S phase and an increasing number of cells in the G1 phase. For all data, representative experiments out of 2 are shown. Each sample was analyzed in triplicate (*n* = 3).

**Figure 5 cancers-13-01094-f005:**
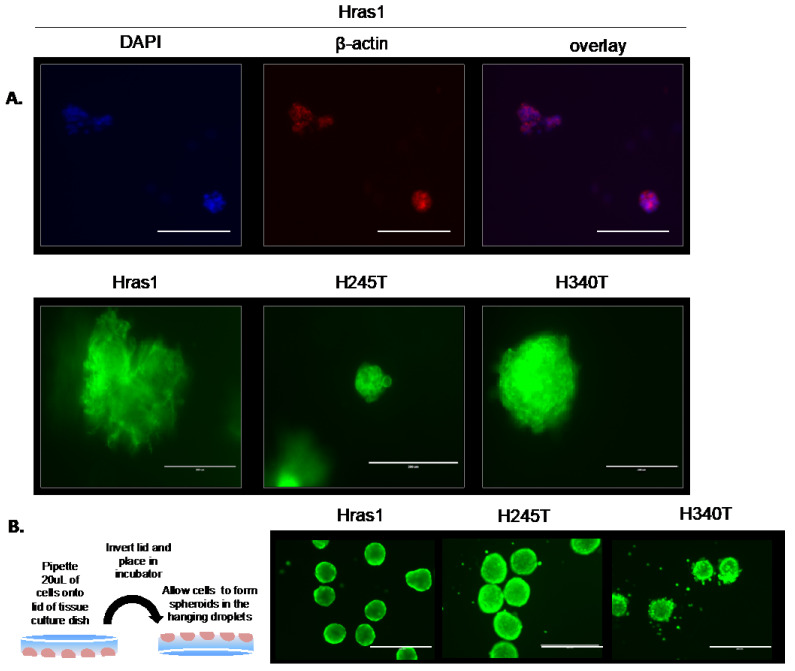
Development of physiologically relevant three-dimensional in vitro tumor models. (**A**) Representative images of *Hras^G12V^/Pten^−/−^/TPO Cre* cell lines (Hras1, H245T, H340T) grown embedded in Matrigel to produce 3D spheroids. Spheroids were fixed and stained for β-actin and counterstained with DAPI (top) or stained with fluorescently conjugated phalloidin antibody (bottom). Scale bars represent 200 µm. (**B**) Left: Schematic depicting process by which spheroids in hanging droplets are generated. Right: Spheroids collected after 7 days were labeled with Calcein-AM to assess viability. Scale bars represent 400 µm. Images are representative of three independent experiments.

**Figure 6 cancers-13-01094-f006:**
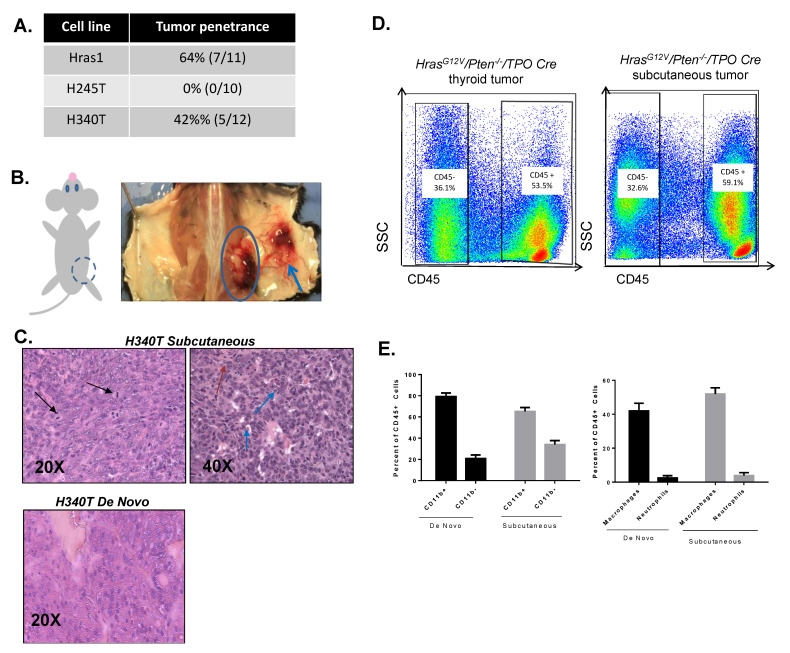
Development of syngeneic tumor model. (**A**) Tumor penetrance rates for all *Hras^G12V^/Pten^−/−^/TPO-Cre* thyroid tumor cell lines (Hras1, H245T, H340T) representing the number of mice with visible or palpable tumors as a percentage of total mice injected. Mice were monitored for 15 weeks, at which point all were sacrificed. Subcutaneous injections were repeated at least three times with each experiment including ten or more recipient mice (*n* = 10–12). (**B**) Gross image of subcutaneous tumor (blue circle) and neovascularization (arrow) on the right hind flank of a wild-type (WT) recipient in comparison to the control contralateral hind flank. (**C**) H&E staining of subcutaneous tumor sections derived from subcutaneous *Hras^G12V^/Pten^−/−^/TPO-Cre* thyroid tumor cells (H340T). Subcutaneous tumors were poorly differentiated or anaplastic with multiple mitotic cells (*top left, black arrows*) in addition to nuclear vesicular changes (*top right, red arrow*) and prominent nucleoli (*top right, blue arrows*). In contrast to H340 source tumor (bottom), subcutaneous tumors demonstrated complete loss of normal follicular pattern. (**D**) Representative FACS analysis of CD45+ immune cell population in de novo *Hras^G12V^/Pten^−/−^/TPO-Cre* thyroid tumor *(left)* and subcutaneous tumor *(right)* derived from an *Hras^G12V^/Pten^−/−^/TPO-Cre* cell line (H340T). Original blots see Appendix A (**E**) FACS analysis of immune cells within the TME of de novo *Hras^G12V^/Pten^−/−^/TPO-Cre* tumors and subcutaneous tumors derived from H340T. Results demonstrate both models are enriched with cells of the myeloid lineage (CD11b+, *left*). Within the myeloid lineage, macrophages represent the largest percentage of cells in the TME (*right*) All groups were compared via two-way ANOVA with Sidak test for multiple comparisons. No statistically significant differences between models were detected.

**Figure 7 cancers-13-01094-f007:**
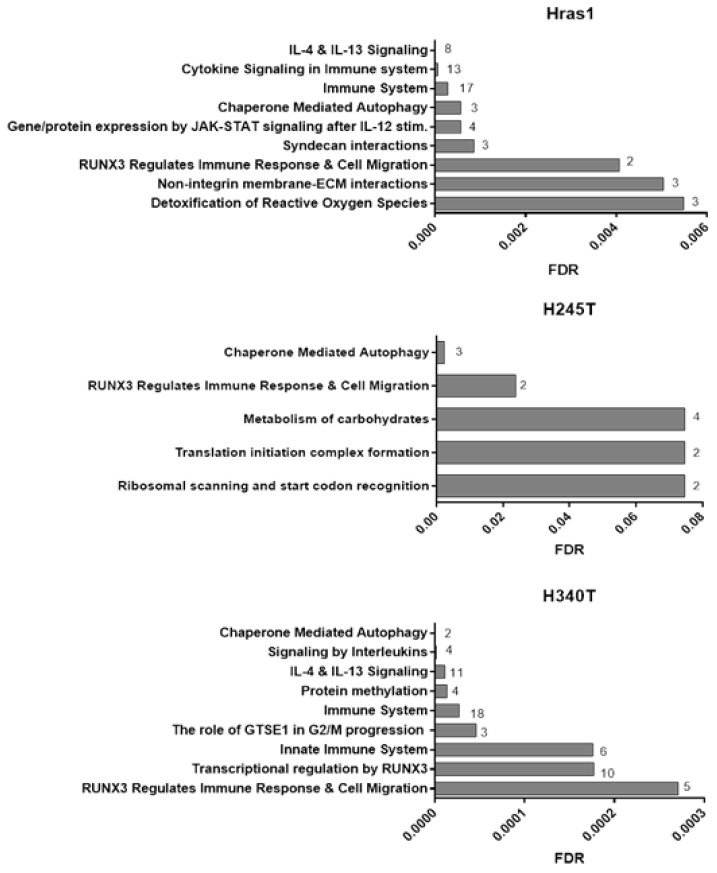
Proteomic analysis of conditioned media from HrasG12V/Pten^−/−^/*TPO-Cre* thyroid cell lines. Pathway analysis results comparing secretions from HrasG12V/Pten^−/−^/*TPO-Cre* thyroid cell lines (Hras, H245T, H340T) and wildtype thyrocytes. Reactome pathway database was queried for all secreted proteins that were statistically different from wildtype (multiplicity adjusted *p*-value of ≤0.05) and demonstrated at least a two-fold increase or decrease in secretion. False discovery rate (FDR) values were calculated using REACTOME software and over-representation analysis to determine significance. Numbers in bar graphs represent the number of proteins found within each pathway. Proteomics analysis was conducted on three independent conditioned media isolations (*n* = 3).

## Data Availability

Data supporting the findings of this study are contained within the article and Appendix A or are available from the corresponding author (A.T.F.) upon reasonable request.

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
