# Peer review of "Development of Novel Follicular Thyroid Cancer Models Which Progress to Poorly Differentiated and Anaplastic Thyroid Cancer"

_cancers, 2021, doi:10.3390/cancers13051094_

Round 1

Reviewer 1 Report

The manuscript describes the generation and characterization of 3 cell line based mouse models of follicular thyroid cancers.  The authors generated 3 cell lines from thyroid cancers in HrasG12V/Pten-/-/TPO-Cre mice.  Thyroid origins of the cell lines were confirmed by deletion of Pten, STR profiling and expression of some thyroid markers.  Dependency on MAPK and PI3K pathways for growth was confirmed with drugs targeting these pathways.  For in vivo characterization, the cell lines were injected into the flank of syngeneic 129SvJ mice.

Comments.

The authors indicated that both FTC and PDTC develop in the HrasG12V/Pten-/-/TPO-Cre mice. It is unclear from data shown whether the cell lines were derived from a PDTC or FTC.  Histology comparing the original tumor to the allograph tumor should be shown.  The histology of the allograph shown in figure 6C appears to be anaplastic.  The spindle shape of cells in culture and loss of EPCAM expression suggest the cells have undergone EMT which is commonly seen in anaplastic thyroid cancers.  Thus, it seems the cell lines are more consistent with a model of anaplastic thyroid cancer than PDTC or FTC.

The cell lines could potentially be an important tool for studying immunotherapies or how targeted therapies change the tumor microenvironment.  For the cell lines to be an informative in vivo model of immunotherapies it is important to understand how well the allograph TME resembles that of the primary tumor. Showing a similar level of Cd45+ cell is not sufficient. 

Other reports have suggested the TME is influenced by the local environment, thus it is unclear why in vivo characterization was done in the flank compared to orthotopic.

The PCR data suggesting metastatic seeding or the presence of circulating tumor cells is missing controls and is not definitive, thus should not be included.

Data showing effects of the MAPK and PI3K inhibitors on their targeted pathways is missing.

Minor comments.

Please provide more details on STR profiling.  Was profiling of the cell lines compared to that of the original tumor/mouse?

Figure 1C.  Internal control is missing.  It would be more informative to perform a multiplex PCR for floxed and recombined Pten allele.

Figure 1D.  The β-actin Western blot the signal is saturated.  Is a different exposure/blot available? 

Since cell lines were injected into 129SvJ mice, I assume the HrasG12V/Pten-/-/TPO-Cre mice are congenic for 129SvJ.  If correct, please add this information to the material and methods.

Reviewer 2 Report

This paper reports an attempt to develop and characterize cell-based models of follicular thyroid cancer that closely mimic the genetic and pathological progression of the disease seen in patients.

The authors have created mice with thyroid-specific expression of HrasG12V and homozygous Pten inactivation [HrasG12V/Ptenhom/TPO-Cre mice]. These mice develop follicular thyroid carcinomas (FTC)s that progress to poorly differentiated thyroid carcinoma (PDTC). The authors have derived cell lines from these thyroids and validated them at the genetic level. However the cell lines, while retaining PAX8 expression, had reduced TTF1 and no expression of TTF2, thyroglobulin or the TSH receptor.

These data are nicely reported and the authors show that they can grow the cell lines in vitro in hanging droplets, and they have developed a syngeneic model in flank implants.

The major problem with this paper is that their model is actually most akin to anaplastic thyroid carcinoma; the cell lines are not at all representative of FTC or even PDTC.  Therefore the interest is limited to the most rare form of thyroid carcinoma. Moreover, PTEN mutations are exceptionally rare in human thyroid carcinomas.  For this reason, the introduction, discussion and abstract should be rewritten to portray the reality of this limited application

Reviewer 3 Report

The authors have developed novel follicular thyroid cancer cell lines from the thyroid tumors of the follicular thyroid cancer mouse model. These cell lines can be utilized in thyroid cancer research depending on how closely they relate to the human FTC. The study has been conducted well; however, future studies are required to establish their relevance in thyroid cancer research. 

Here are the comments to the authors:

  1. The authors have not provided information on how many times the in vitro and in vivo experiments have been repeated and the number of replicates (n) in each experiment.
  2. The authors should provide STR analysis information in the article.
  3. The authors should have also performed a sorting experiment to enrich the cell lines for thyroid cancer cells and reduce any stromal cell contamination. 

Round 2

Reviewer 1 Report

The additional data and clarification provided by the authors addresses primary concerns of this reviewer.

Thank you.